# GPBSO: Gene Pool-Based Brain Storm Optimization for SNP Epistasis Detection

**DOI:** 10.3390/genes16091114

**Published:** 2025-09-19

**Authors:** Liyan Sun, Yi Xin, Shen Qu, Linxuan Zheng, Linqing Jiang

**Affiliations:** 1School of Computer Science and Technology, Changchun University, Changchun 130022, China; sunly@ccu.edu.cn (L.S.); zlx20010226@163.com (L.Z.); taixvhuanjing@163.com (L.J.); 2School of Mathematics and Statistics, Changchun University, Changchun 130022, China; qs1779936008@163.com

**Keywords:** epistasis, genome-wide association studies, single-nucleotide polymorphism

## Abstract

**Background/Objectives:** Detecting high-order epistatic interactions in genome-wide association studies (GWAS) is essential for understanding complex diseases, yet most existing approaches are limited to pairwise interactions. We propose GPBSO (Gene Pool-Based Brain Storm Optimization for Epistasis Detection), a novel stochastic framework that integrates Brain Storm Optimization with a dynamic gene pool to efficiently explore high-order SNP combinations. **Methods:** Epistasis is evaluated using the k2 Bayesian network scoring criterion and the G-test, with iterative updates to the gene matrix enhancing search diversity. **Results:** Comparative experiments on simulated datasets generated from five epistatic models demonstrated that GPBSO consistently outperformed a set of well-established methods—DECMDR, SNPHarvester, AntEpiSeeker, HS-MMGKG, and SEE—in terms of F-measure and statistical power, particularly for third-order interactions. **Conclusions:** GPBSO provides an effective and scalable approach for detecting high-order epistatic interactions, offering methodological advancements for genetic epidemiology and complex disease analysis.

## 1. Introduction

With the rapid advancement of extremely high-capacity sequencing technologies, researchers can now readily examine vast numbers of SNP markers—often reaching into the hundreds of thousands—within datasets containing thousands of participants [1,2]. Against this backdrop, genome-wide association studies (GWAS) [3,4,5,6,7,8] have emerged as the principal paradigm for dissecting the genetic architecture of common, polygenic disorders. These investigations seek to uncover statistical dependencies between allelic variation and disease status by systematically interrogating dense SNP panels in large case–control cohorts. Contemporary GWAS datasets typically encompass thousands of phenotypically ascertained subjects and upwards of 1 × 10^5^–1 × 10^6^ genotyped variants, and have already yielded an extensive catalogue of phenotype-associated loci [9,10,11,12,13,14].

Nevertheless, the etiological interpretation of complex traits necessitates an explicit consideration of non-additive, genome-wide epistatic interactions. Epistasis-defined as the context-dependent effect of one locus contingent upon the allelic state of another [15,16,17,18,19,20]—is increasingly recognized as a pivotal component of genetic architecture. Under this model, individual susceptibility alleles exert negligible marginal effects, yet collectively generate pronounced phenotypic perturbations via synergistic or antagonistic interactions. Consequently, exhaustive detection of such combinatorial effects is indispensable, but introduces formidable computational complexity due to the exponential expansion of the multi-locus search space.

Over the past decade, a diverse portfolio of computational strategies has been introduced to systematically uncover epistatic interactions [21,22,23,24,25,26,27,28,29,30]. Among these, Multifactor Dimensionality Reduction (MDR) [21] pioneered a non-parametric combinatorial framework that collapses high-dimensional genotype profiles into discrete risk classes, thereby facilitating the detection of multilocus associations. Its original formulation, however, exhibits quadratic complexity in the number of single-nucleotide polymorphisms (SNPs), restricting its applicability to datasets comprising only tens of variants.

To mitigate this scalability bottleneck, DECMDR [22] hybridises Classification-based MDR (CMDR) with Differential Evolution (DE), employing CMDR-derived classification accuracy as the fitness landscape upon which DE navigates large-scale GWAS data. Stochastic search has been further advanced by SNPHarvester [24], which leverages probabilistic pruning to discard loci unlikely to participate in epistatic effects, thereby dramatically compressing the search space.

Multi-objective optimisation paradigms have also gained prominence. MACOED [25] integrates mutually complementary objective functions derived from logistic regression and Bayesian network scoring within a memory-enhanced Ant Colony Optimisation (ACO) [28] architecture, enabling simultaneous maximisation of predictive power and model parsimony. AntEpiSeeker [26] adopts a dual-stage ACO procedure: an initial coarse-grained exploration of moderately sized SNP subsets followed by an exhaustive examination of the most pheromone-enriched loci. Harmony-search-based HS-MMGKG [29] expands the objective space to five criteria, while SEE [30] implements an eight-objective evolutionary algorithm—four conventional association metrics augmented by four measures quantifying the marginal contribution of each SNP within a candidate combination.

Despite these algorithmic advances, prevailing approaches exhibit several persistent limitations. Most methods mandate an a priori specification of interaction order, struggle to maintain an optimal balance between stochastic exploration and population diversity, and lack unified, high-resolution evaluation criteria capable of rigorously quantifying the biological significance of detected epistatic effects.

Herein, we present GPBSO—Gene Pool-Based Brain Storm Optimization—a new swarm-intelligence algorithm devised for the systematic discovery of epistatic interactions. Distinct from prevailing approaches, GPBSO integrates automatic estimation of the maximum interaction order, the exploratory power of a dynamically maintained gene pool, and the balance between global and local search achieved by the Brain Storm Optimization framework. In addition, GPBSO combines Bayesian network scoring with statistical validation to ensure robust detection of epistatic interactions across multiple orders. These innovations collectively make GPBSO a scalable and reliable tool for dissecting the polygenic architecture of complex traits.

To thoroughly evaluate the effectiveness of GPBSO, we employed both synthetic and real-world GWAS datasets. The synthetic datasets were generated using EpiGEN [31], a simulation tool that produces genotype data with realistic genetic characteristics, including deviations from Hardy–Weinberg equilibrium, linkage disequilibrium patterns, and user-specified minor allele frequencies. Five distinct epistatic interaction models—joint-recessive, joint-dominant, modular, diagonal, and XOR. These datasets enabled a systematic evaluation of GPBSO’s performance, which was benchmarked against five existing epistasis detection methods [22,24,26,29,30].

Subsequently, GPBSO was deployed on the Wellcome Trust Case Control Consortium (WTCCC) GWAS dataset [32]. The algorithm recovered epistatic networks encompassing a broad spectrum of interaction orders. A subset of implicated genes—*TSBP1*, *TSBP1-AS1*, *DAB1*, *OMA1*, *STK32A-AS1*, *LRIG1*, *LOC105376945*, *BTNL2*, *SPATA5*, and *LOC105370777*—exhibited corroborative evidence in the Comparative Toxicogenomics Database (CTD) [33]. Visualization of SNP-level and gene-level interaction networks revealed previously uncharacterized loci that may contribute to the etiology of the seven complex diseases investigated by the WTCCC. Collectively, these findings establish GPBSO as a robust and scalable approach for dissecting the polygenic architecture of common human diseases.

## 2. Materials and Methods

When faced with a difficult problem that no single individual can solve, bringing together a group of people—especially those with diverse backgrounds—for brainstorming often leads to a high probability of finding a solution. Human interaction and collaboration can generate remarkable and unexpected intelligence. One effective way to facilitate such interaction and cooperation for generating innovative ideas is to organize a brainstorming session. This concept led to the invention of Brain Storm Optimization (BSO) [34]. Since its inception, the BSO algorithm has garnered significant attention in the field of swarm intelligence research.

The Gene Pool-based Brain Storm Optimization (GPBSO) algorithm is an improved method based on BSO, designed to detect complex epistatic interactions in Genome-Wide Association Studies (GWAS) data. GPBSO enhances the search process by emulating the collaborative and ideation dynamics characteristic of human brainstorming, while simultaneously employing a dynamic gene pool management strategy. Initially, the gene pool is defined as the complete set of SNPs not yet incorporated into the existing population. As the algorithm progresses, this pool is adaptively modified: when a newly generated individual is admitted into the population, its constituent SNPs are excluded from the gene pool; conversely, when an individual is eliminated due to suboptimal fitness, its SNPs are returned to the pool. This continuous update mechanism ensures that the gene pool remains representative of the SNPs available for future selection, thereby sustaining population diversity and maintaining an effective balance between exploration and exploitation. The structural overview and procedural workflow of GPBSO are illustrated in Figure 1, with detailed explanations of each component presented in the subsequent sections.

### 2.1. Load the Gwas Data into Memory

To enhance computational efficiency and minimize memory overhead, the algorithm utilizes a binary-based data storage and processing strategy, analogous to the method implemented in BOOST. Given a dataset comprising N genetic loci, along with C0 control samples and C1 case samples, the genotype information is organized into two distinct data structures: healthy_data and disease_data, corresponding to the control and case groups, respectively. The healthy_data structure is defined as an array of length N, where each entry healthy_data[loc] (with loc in the range [0, N)) is a vector consisting of three components representing the encoded genotype counts or distributions for the respective locus. Each component is a binary array of length C0, representing the genotype distribution of the loc-th gene locus in the healthy group. Specifically, healthy_data[loc][type][sample] (sample ∈ [0, C0)) takes a value of 1 or 0, indicating whether the genotype of the sample-th healthy sample at the loc-th locus matches the type-th possible genotype. The disease_data array follows the same structure as healthy_data. By using only 3 bits per gene locus and sample to store genotype information, the algorithm significantly reduces memory usage. For example, if each gene locus has two possible genotypes (0 or 1) and the dataset contains 8 disease samples and 8 healthy samples, a statistical table for gene locus combinations is constructed through binary operations, which is then used to calculate relevant statistical metrics. The binary operations on genotype data significantly reduce computational complexity, with a time complexity of O(S × L × 3^L^), where L is the number of gene loci in the combination and S is the total number of samples. Despite the seemingly high complexity, binary logic-based operations ensure efficient execution. Previous studies, such as HS-MMGKG [29], SEE [30], and BOOST [35], have demonstrated that this storage structure and binary operations can significantly improve computational performance.

### 2.2. Determine the Maximum Order (mo) Based on the Samples

GPBSO does not require a preset interaction order. Instead, it determines the maximum order (mo) based on the GWAS sample size. The maximum interaction order mo is estimated based on the smaller of the case or control sample sizes. This estimation involves computing the natural logarithm of the smaller sample size, subtracting 0.5, and then rounding the result down to the nearest integer. While a larger mo enables the algorithm to investigate more complex epistatic interactions, setting mo too high may result in excessively sparse contingency tables, thereby compromising the reliability of the k2 statistic. To ensure analytical stability, the algorithm is designed such that each cell in the contingency table contains approximately three samples, with the final mo value adjusted accordingly through rounding.

### 2.3. Initialize Discussion Groups and Gene Pool

The initialization of discussion groups and the gene pool constitutes a critical component of the GPBSO algorithm, directly influencing its operational efficiency and optimization capability. At the initial stage, the algorithm employs the k-means++ clustering method to partition the population into multiple discussion groups. Each group represents a localized region within the solution space and comprises individuals with similar genetic characteristics. This grouping mechanism emulates the collaborative nature of brainstorming sessions, fostering inter-group diversity and intra-group cohesion. Such a structure not only facilitates organized generation of new individuals in subsequent phases but also reinforces the algorithm’s capacity to navigate and exploit the solution space effectively.

Simultaneously, the gene pool module is systematically constructed to complement this framework. It is initialized as the set of all candidate SNPs that are not currently present in any individual within the population. Thus, the gene pool serves as a dynamic repository of unused genetic elements, ensuring access to a wide range of variation during the search process. As the algorithm progresses, the gene pool is continuously updated in real time. When a newly generated individual is successfully incorporated into the population, its constituent SNPs are removed from the gene pool to prevent repeated usage of the same genetic components, thereby enhancing population diversity and encouraging exploration of unexplored regions in the search space. Conversely, if an individual is removed due to inferior fitness, its SNPs are restored to the gene pool, ensuring that potentially useful genetic material remains accessible.

The coordinated implementation of discussion group formation and dynamic gene pool management enables GPBSO to maintain a robust balance between exploration and exploitation. This balance is essential for mitigating the risk of premature convergence and promoting an efficient, non-redundant search process. Collectively, the design and integration of these two modules represent key innovations in the algorithm, substantially contributing to its overall performance and effectiveness in identifying epistatic interactions. The operational structure of the gene pool module is depicted in Figure 2.

### 2.4. Generate Individual X Through Discussion Group Dynamics

The individual generation algorithm in the GPBSO algorithm embodies the core principles of the framework, which dynamically balances the exploitation of promising ideas and the exploration of new ones by simulating brainstorming sessions. Utilizing three distinct methods, the algorithm mimics the collaborative and innovative nature of brainstorming, leveraging the collective intelligence of “discussion groups” and the untapped potential within the gene pool.

In the Idea Fusion Method, a top-performing discussion group is selected to refine existing ideas. Two high-ranking individuals (ideas) from this group are chosen, and their genetic material is combined—taking the first half of the SNPs from one and the second half from the other. This reflects the brainstorming process where participants merge the best aspects of existing ideas to create a stronger, more refined solution. This strategy enables the algorithm to intensify its search within promising regions of the solution space, leveraging the advantages of previously validated solutions.

Within the Inspiration Sparking Method, the process of partially reimagining an idea based on external stimuli is emulated. Specifically, a high-fitness individual selected from a designated discussion group undergoes targeted modification, wherein the second half of its SNP sequence is substituted with novel SNPs sampled from the gene pool. This mechanism introduces controlled variability while preserving the core attributes of the original solution, thereby achieving a balance between exploiting established knowledge and exploring new combinatorial possibilities.

The Free Association Method reflects the brainstorming principle of unconstrained ideation by generating entirely novel individuals. In this approach, a new candidate is constructed solely from the gene pool by randomly selecting a fixed number of SNPs. This encourages the incorporation of original and independent genetic combinations into the population, thereby enhancing solution diversity and reducing the risk of premature convergence to local optima.

Collectively, these three generation strategies embody the fundamental tenets of brainstorming—structured synthesis, adaptive transformation, and radical innovation. The Idea Fusion Method facilitates refinement through recombination of existing individuals, the Inspiration Sparking Method promotes guided creativity via selective alteration, and the Free Association Method ensures exploratory breadth through the introduction of wholly new genetic material. The interplay of these mechanisms enables the algorithm to thoroughly and adaptively traverse the search space, enhancing its ability to identify intricate patterns of SNP epistasis. The coordinated function of discussion groups and the dynamically managed gene pool further contributes to the algorithm’s capacity to balance intensification and diversification during the search process.

### 2.5. Identifying Epistasis Within SNP Combinations Through the k2 Metric

The k2 score, which is commonly used in Bayesian network modeling, serves as a quantitative indicator for detecting epistatic associations. Its formal definition is shown in Equation (1).(1)$$k2(Y,S)=\prod_{c\in C}\frac{m_{c,0}!\times m_{c,1}!}{(m_{c,*}+1)!}$$

In Equation (2), the k2 statistic quantifies the degree of association between phenotype traits and a group of SNP interactions. *S* refers to a specific combination of SNPs, and *Y* represents the phenotype under investigation. The set *C* contains all possible genotype configurations for *S*—for instance, if there is one SNP, then *C* includes three such patterns. For each genotype *c* in *C*, the term *m_c_*_,*_ captures the total number of samples exhibiting that genotype. More specifically, *m_c_*_,0_ and *m_c_*_,1_ refer to the number of control and case samples, respectively, for genotype c.

The k2(*Y*, *S*) score quantifies the goodness-of-fit of a Bayesian network with phenotype Y and SNP set S. Lower scores denote sparser, more accurate models and thus stronger associations. Iteratively withholding each SNP x ∈ S and re-evaluating k2(Y, S{x}) reveals the variable’s contribution: removal of an uninformative SNP tightens the model and lowers the score, whereas excising a relevant or epistatic SNP inflates it. GPBSO exploits this sensitivity by systematically pruning SNPs until no deletion further reduces k2(*Y*, *S*). If the final retained subset exceeds two members, it is declared an epistatic interaction (Algorithm 1).
**Algorithm 1** Identify Epistatic Interactions Using k2 Metric**Require:** Maximum order *mo*; SNP combination *x* with *mo* SNPs generated by GPBSO; *k2x* as the *k2* score of *x***Ensure:** Epistatic interaction of order within [2, *mo*) or no result1:Set *l* = *mo*2:**while** *l* > 1 **do**3: *foundImprovement* = false4: **for** each SNP index *i* in [0, *l*) **do**5:  Create a new SNP combination *xx* of length *l* - 1 by excluding *x*[*i*]6:  Compute *k2xx* as the *k2* score for *xx*7:**  if** *k2xx* < *k2x* **then**8:   Update x to xx9:   Update k2x to k2xx10:   Decrement *l* by 111:   Set *foundImprovement* = true12:    Exit the loop13:**  end if**14:** end for**15: **if** not *foundImprovement* **then**16:  Exit the loop17:** end if**18:**end while**19:**if** *l* > 1 **then**20:** return** *x* as the detected epistatic interaction21:**end if**

### 2.6. Evaluating the Significance of Interaction Effects with the G-Test

Within the GPBSO framework, epistatic relationships are evaluated and categorized as either statistically relevant or not. As illustrated in Algorithm 1 the algorithm can uncover a large number of such interactions. This section outlines how GPBSO determines statistical significance, using the G-test as its core evaluation method. The G-test [36], grounded in likelihood ratios and maximum likelihood estimation, is widely used for hypothesis testing. Under the assumption of no association with the phenotype, the G-statistic approximately follows a chi-squared distribution, which supports its suitability for detecting true epistatic signals. In our analysis, the G-test *p*-value (denoted as g) [30] is used to evaluate significance. The equation used to compute g is presented in Equation (2) on the following page. An interaction is marked as significant when g falls below a user-defined threshold, in which case it is included in the final output file.

In Equation (2), the statistical significance of the relationship between a given phenotype and a specific set of SNPs is measured using the G-test. This evaluation is carried out under the likelihood-ratio framework, and its test statistic approximately follows a chi-squared distribution. All possible genotype configurations for the chosen SNP set are considered. For example, if the set contains l SNPs, there will be 3^l^ potential genotype patterns.(2)$$g(Y;S)=\mathrm{pvalue}\_\mathrm{of}(2\sum_{c\in C}m_{c,0}\times \frac{m_{c,0}\times m}{m_{c,*}\times m_{*,0}}+m_{c,1}\times \frac{m_{c,1}\times m}{m_{c,*}\times m_{*,1}})$$

Here, *m* refers to the total number of samples. For a particular genotype *c* in *C*, *m_c_*_,0_ and *m_c_*_,1_ indicate the number of control and case samples, respectively. The total number of individuals with genotype c is given by *m_c_*_,*_ Across all genotype combinations, m_*,0_ and m_*,1_ represent the total number of control and case samples in the entire dataset. Finally, min *g*(*Y*; *E*) corresponds to the smallest *p*-value among the SNPs in *S*.

### 2.7. Parameter Settings and Practical Guidance

The performance of GPBSO depends on three key parameters: the population size (PopSize), the maximum number of generations (maxGen), and the maximum interaction order (mo). In all experiments, mo was estimated automatically from the sample size as described in Section 2. By contrast, PopSize and maxGen are user-specified and should be chosen according to dataset size, targeted interaction order, and available compute. Across our simulated scenarios, PopSize ranged from 50–200, while maxGen had no prescribed minimum and was set by the user; in practice, values between 10^6^ and 8 × 10^7^ worked well for the scales studied. For the WTCCC analyses, we adopted automatic order inference (o = −1), maxGen = 8 × 10^7^, and a fixed random seed (0). These settings provided a practical balance between runtime and detection power.

For practical applications, we recommend setting PopSize = 50–200. Smaller datasets (<500 SNPs) can be analyzed with PopSize = 50–100 and a modest maxGen (often around 10^6^–10^7^, but smaller values are permissible if convergence is adequate), whereas larger panels (>10,000 SNPs) benefit from PopSize = 150–200 and larger maxGen (up to 8 × 10^7^ in our experiments). When compute is limited, first cap mo at 2–3 and/or reduce PopSize, then increase maxGen gradually until performance metrics (e.g., F-measure/power) stabilize across independent runs. To enhance usability and reproducibility for other researchers, we provide the parameter settings of the algorithms in Appendix A.

## 3. Results

### 3.1. Performance Assessment Using Simulated Genotype Data

There are a number of challenges related to generating simulated epistasis datasets. From a biological perspective, in real genetic environments, alleles tend to only partially conform to Hardy–Weinberg Equilibrium, which is influenced by the Minor Allele Frequency (MAF), and are locally constrained by linkage disequilibrium structures. Ultimately, any interaction that contributes to the causation of a phenotype may explain only part of the trait’s heritability, leaving a substantial portion unaccounted for. Additional practical challenges involve runtime, computational efficiency, and scalability. To systematically evaluate the performance of GPBSO and competing algorithms on these simulated datasets, we employed two commonly used metrics: F-measure, defined as the harmonic mean of precision and recall, and Power, defined as the probability of correctly identifying true epistatic interactions.

To address these complexities, we employed EpiGEN to generate simulation datasets that reflect realistic genetic properties. EpiGEN was used to generate a corpus of synthetic genetic data based on 2000 samples modeled on Chromosome 22. Several models were constructed based on potential epistatic interactions described by Evans et al. [37], including joint-recessive, joint-dominant, modular, diagonal, and XOR interaction types (Table 1). These models were achieved by adjusting the case–control ratios of the involved genotypes to modify penetrance. The penetrance table for third-order interactions is provided in Appendix A. Exploratory data analysis was used to identify a range of ratios that distinguished the performance of the evaluated tools. To increase feature complexity, up to 500 noise loci were added, and the Minor Allele Frequencies (MAFs) were constrained between 0.05 and 0.5. Each dataset was targeted to include 1000 cases and 1000 controls, although the actual counts varied slightly to accommodate modeling requirements.

We constructed three types of simulation scenarios: (1) second-order interactions among 100 SNPs, (2) second-order interactions among 1000 SNPs, and (3) third-order interactions among 100 SNPs. In the third-order interaction scenario, the 100 loci included 3 interacting loci and 97 noise loci. For each scenario, ten independent datasets were generated for each of the five epistatic interaction models, resulting in a total of 150 simulated datasets.

Using these datasets, we systematically evaluated the performance of GPBSO and benchmarked it against five existing methods [22,24,26,29,30], under all three simulation scenarios. All six software tools and their parameter configurations are summarized in Appendix A. Among them, DECMDR, HS-MMGKG, and SEE are capable of detecting epistatic interactions of any order, whereas SNPHarvester and AntEpiSeeker are limited to second-order interactions. Notably, AntEpiSeeker failed to process third-order interactions due to a segment fault.

To visually illustrate the detection performance of each method, Figure 3 presents six labeled panels (a–f), showing the distributions of F-measure and power for three simulation scenarios: 2nd-order epistasis with 100 SNPs (a,b), 2nd-order epistasis with 1000 SNPs (c,d), and 3rd-order epistasis with 100 SNPs (e,f). Panels (a–d) compare GPBSO with five existing algorithms, resulting in six boxplots per panel, while panels (e,f) compare GPBSO with three algorithms, resulting in four boxplots per panel. Each panel summarizes results aggregated across five distinct epistatic interaction models, each based on ten independently generated datasets. Detailed per-model results are provided in Appendix A. Overall, GPBSO achieves superior performance compared with competing methods across different simulation settings.

### 3.2. Computational Cost Analysis

To further evaluate the efficiency of GPBSO, we compared its computational cost with that of competing algorithms under the same experimental environment. Table 2 summarizes the average runtime (in minutes) across datasets of varying SNP sizes. The results indicate that GPBSO requires more time than filtering-based approaches such as SNPHarvester, but achieves comparable or lower runtime than other swarm intelligence and evolutionary algorithms, while maintaining superior detection power. These findings demonstrate that GPBSO strikes a balance between efficiency and accuracy, and that its runtime remains practical for both simulated and real GWAS datasets.

### 3.3. Validation on Authentic GWAS Datasets

Real-data validation used the Wellcome Trust Case Control Consortium (WTCCC) resource: 14,000 patients across seven complex diseases and 3000 shared controls. The disorders are bipolar disorder, coronary artery disease, Crohn’s disease, hypertension, rheumatoid arthritis, type 1 diabetes, and type 2 diabetes, each represented by about 2000 cases. All samples were genotyped for approximately 500,000 SNPs. Following WTCCC quality-control guidelines, we removed low-quality SNPs and samples and excluded monomorphic loci, yielding seven independent case–control GWAS datasets; details are in Appendix A.

Building on the favorable outcomes observed in the simulation phase, we extended GPBSO to the seven WTCCC GWAS datasets. Runs were configured with automatic order inference (o = −1), a generation cap of 8 × 10^7^, a fixed random seed (0), and all remaining parameters at their defaults. Representative epistatic interactions are reported in Table 3; the full compilation is provided as Appendix A.

Leveraging annotations from the NCBI database of genetic variation (dbSNP) [38], each associated SNP was mapped to its corresponding gene(s). Gene and gene-pair frequencies were subsequently tallied and are displayed in Table 4 and Table 5, respectively. Entities exhibiting elevated occurrence counts are highlighted as candidate drivers of disease susceptibility. To further evaluate the biological relevance of these candidates, the implicated genes were cross-referenced against the Comparative Toxicogenomics Database (CTD), which provides curated evidence linking genes to human diseases. Several of the top-ranked genes, such as *DAB1*, *BTNL2*, and *LRIG1*, are documented in CTD as being associated with neurological, immune, or proliferative disorders, consistent with their potential roles in the diseases analyzed here. Others, labeled as “NF” (Not Found), represent novel candidates whose biological functions remain to be clarified in future studies. This cross-validation using CTD adds external support to our findings and enhances their biological interpretability.

To visualize the resulting interaction landscapes, Cytoscape v3.6.0, an open-source platform for biological network visualization [39], was employed to construct disease-specific SNP-level and gene-level networks. Illustrative examples for Bipolar Disorder are provided in Figure 4 and Figure 5; analogous networks for the remaining six diseases are available in Appendix A.

## 4. Discussion

This study evaluated GPBSO for detecting epistatic interactions in GWAS data using both simulated and real datasets. In real GWAS analyses of seven complex diseases, GPBSO identified numerous significant SNP–SNP and gene–gene interactions, many consistent with known biology, supporting the method’s practical utility. Although GPBSO’s computational cost is higher than purely filtering-based methods, its superior performance in detecting higher-order interactions compensates for this limitation. Functional validation remains necessary to confirm the causal roles of detected interactions. In addition to rediscovering known disease-associated loci, GPBSO also revealed several novel candidate genes, including *LRIG1*, *OMA1*, and *SPATA5*. Prior studies suggest that *LRIG1* is involved in cell proliferation and immune signaling [40,41], *OMA1* contributes to mitochondrial homeostasis [42,43], and *SPATA5* is linked to mitochondrial dynamics. These functional insights provide biological plausibility for the novel signals detected by GPBSO, thereby strengthening the interpretability of the real-data application.

Although GPBSO involves several tunable parameters, the provided guidelines can assist researchers in selecting appropriate values according to dataset size and available computational resources, thereby improving reproducibility and usability for other researchers. Future work will focus on developing adaptive strategies for automatic parameter tuning, further optimizing computational efficiency, and integrating biological prior knowledge to enhance interpretability and scalability.

Compared with existing approaches, GPBSO demonstrates several distinctive advantages. First, it eliminates the need for manual specification of interaction order by automatically estimating the maximum order (mo) from the dataset, thereby supporting a comprehensive and scalable search. Second, the Brain Storm Optimization framework enables a robust balance between exploration and exploitation, which helps maintain population diversity and reduces the risk of premature convergence. Third, the incorporation of a dynamically replenished gene pool broadens the search space, enhances heterogeneity, and reduces selection bias in SNP sampling. Finally, the integration of the Bayesian K2 score with the G-test ensures statistically rigorous evaluation of candidate interactions across all orders. Together, these features distinguish GPBSO from other epistasis detection algorithms and account for its superior performance on both simulated and real GWAS datasets.

## 5. Conclusions

In conclusion, our extensive experiments demonstrate that GPBSO is a powerful and practical algorithm for detecting epistatic interactions in GWAS data, with particular strength in uncovering higher-order interactions. Applied to seven real GWAS datasets of complex diseases, GPBSO successfully identified numerous biologically meaningful SNP–SNP and gene–gene interactions, and constructed interaction networks that highlight potential candidate genes for further study. These results not only validate the effectiveness of GPBSO but also provide valuable insights into the genetic architecture of complex diseases. Future work will focus on further improving computational efficiency and integrating functional validation to enhance biological interpretability.

## 6. Patents

This research resulted in a patent entitled “Method for Detecting SNP Epistasis Based on Brain Storm Optimization Algorithm” (China Patent No. CN202410995721.4, filed 24 July 2024, granted 17 September 2024).

## Figures and Tables

**Figure 1 genes-16-01114-f001:**
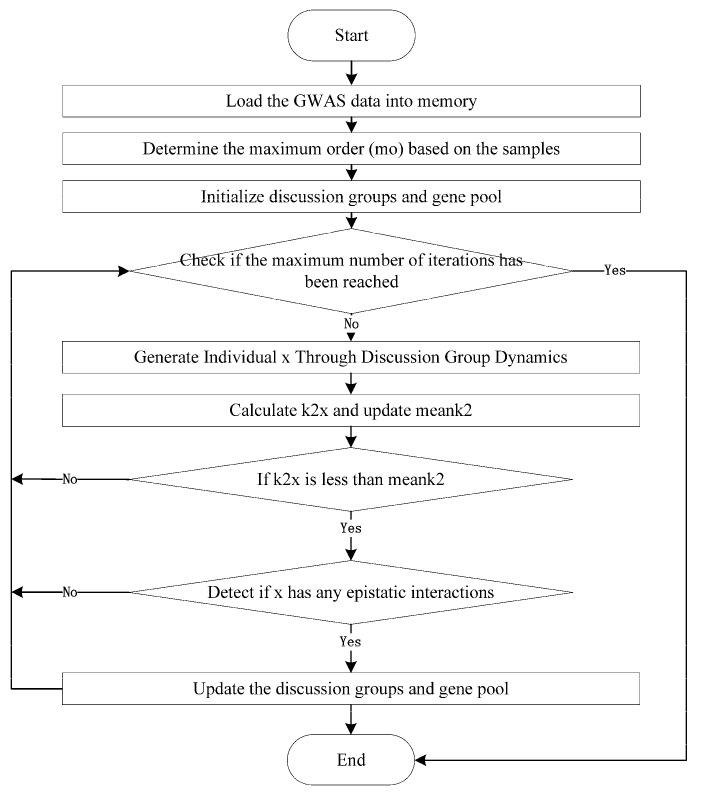
The Pseudo Code of GPBSO Algorithm.

**Figure 2 genes-16-01114-f002:**
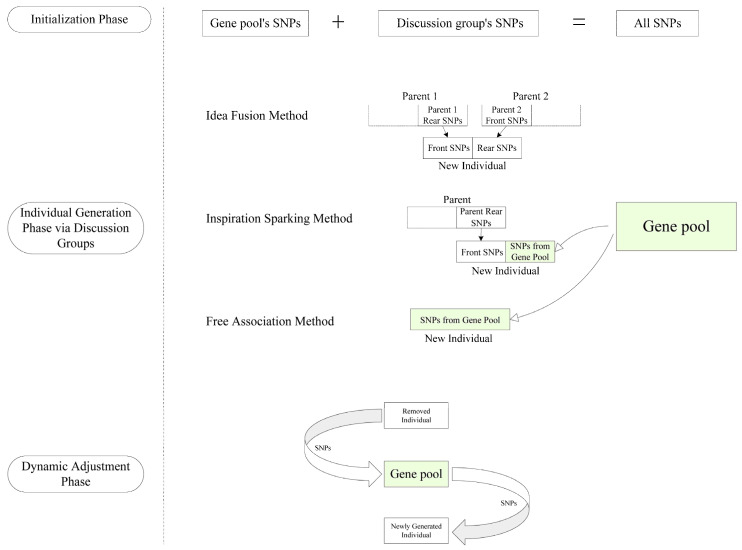
Dynamic Flow Diagram of the Gene Pool.

**Figure 3 genes-16-01114-f003:**
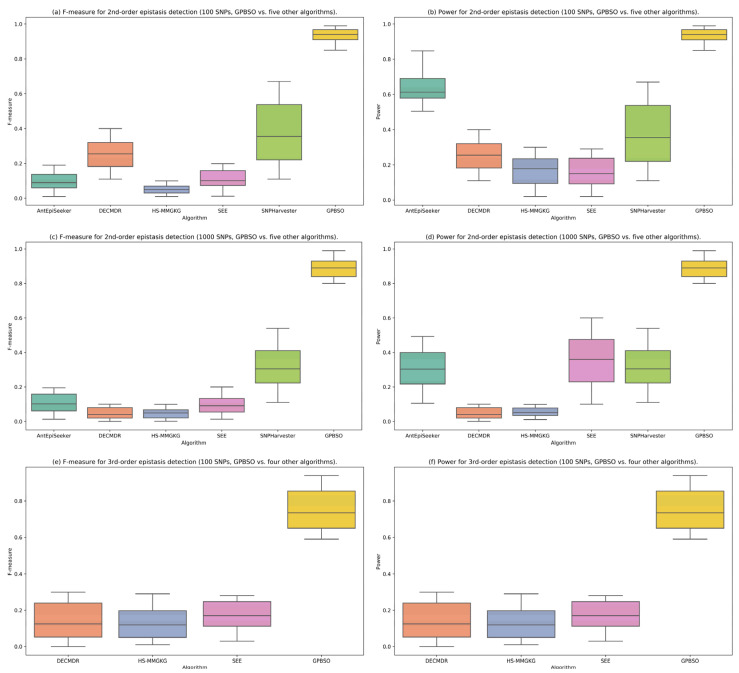
Comparative performance of GPBSO and existing methods on simulated datasets. Panels (**a**–**d**) show the distributions of F-measure and power for 2nd-order epistasis with 100 and 1000 SNPs (GPBSO vs. five other algorithms, six boxplots per panel). Panels (**e**,**f**) present the results for 3rd-order epistasis with 100 SNPs (GPBSO vs. three other algorithms, four boxplots per panel). Each panel summarizes results aggregated across five distinct epistatic interaction models, each based on ten independently generated datasets. These results demonstrate that GPBSO consistently outperforms competing methods in both F-measure and power.

**Figure 4 genes-16-01114-f004:**
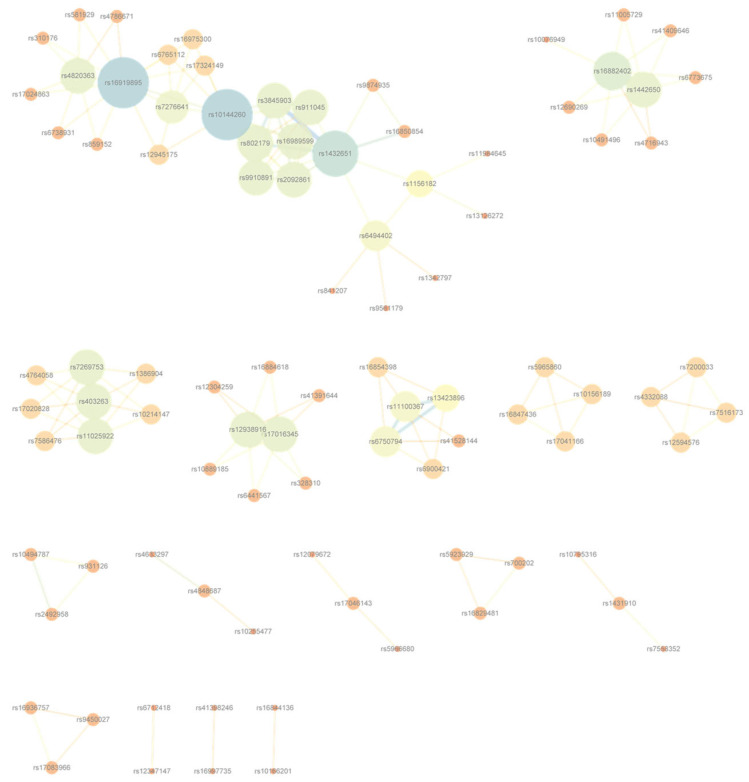
SNP-level interaction network for Bipolar Disorder, constructed from high-confidence SNP pairs with the lowest G-test *p*-values after frequency filtering. Corresponding networks for the remaining six diseases are presented in Appendix A.

**Figure 5 genes-16-01114-f005:**
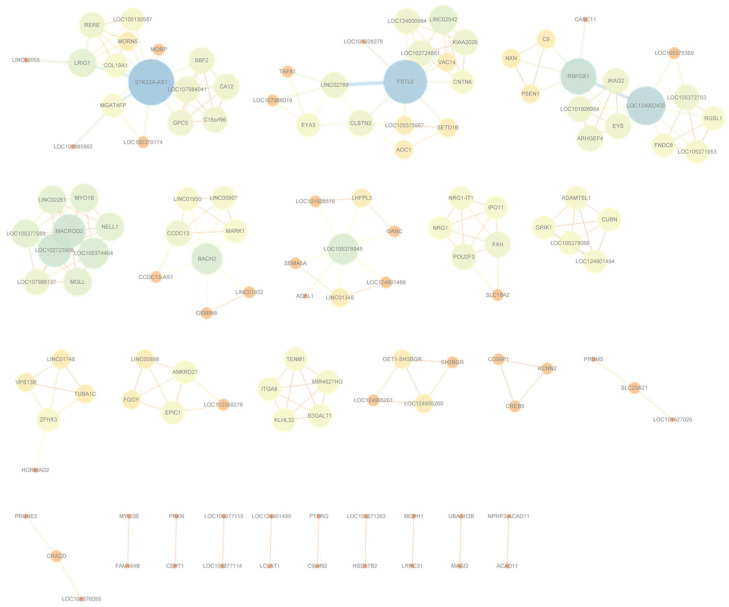
Gene-level interaction network for Bipolar Disorder, constructed from high-confidence gene pairs with the highest co-occurrence frequency after frequency filtering. The corresponding networks for the remaining six diseases can be found in Appendix A.

**Table 1 genes-16-01114-t001:** EpiGEN Penetrance models with capital genotypes as the major allele.

Model	SNP2	AA	Aa	aa
Joint Dominant	BB	0	0	0
	Bb	0	0	1
	bb	0	1	1
Joint Recessive	BB	0	0	0
	Bb	0	0	0
	bb	0	0	1
Modular	BB	0	0	0
	Bb	0	0	1
	bb	1	1	1
Diagonal	BB	1	0	0
	Bb	0	1	0
	bb	0	0	1
XOR	BB	0	0	1
	Bb	0	0	1
	bb	1	1	0

**Table 2 genes-16-01114-t002:** Average runtime (in minutes) of GPBSO and competing algorithms under different simulated dataset sizes.

Method	Dataset	Dataset	Dataset
GPBSO	3	39	25.75
DECMDR	9.5	860	42.5
SNPHarvester	2.25	22.5	not supported
AntEpiSeeker	3.75	50	Fail
HS-MMGKG	1.15	5	3
SEE	2.5	47.25	40

**Table 3 genes-16-01114-t003:** Representative epistatic interactions identified by GPBSO across seven WTCCC GWAS datasets (complete list in Appendix A).

g	snp1	snp2	snp3	snp4
Bipolar Disorder				
0	rs10494787	rs12050604		
0	rs10494787	rs1156182	rs4257642	
0	rs10494787	rs2492958	rs17105919	
0	rs10494787	rs11100367	rs16898339	rs16954052
0	rs16824455	rs355631	rs10494787	rs1442650
0	rs10494787	rs7593135	rs17064749	
Coronary Artery Disease				
0	rs1091486	rs6643915		
0	rs1563315	rs630967		
0	rs5961751	rs4543670		
0	rs6634416	rs4543670		
0	rs6631847	rs4543670		
0	rs646221	rs5963481	rs1412333	
Crohn’s Disease				
0	rs11209026	rs17096872	rs7929346	
0	rs13126272	rs11936062	rs2173972	rs7742263
0	rs11887827	rs17020502	rs1489795	rs7754204
0	rs11887827	rs6752979	rs41321049	
0	rs17083420	rs10968251	rs10144243	
0	rs11209026	rs10045484	rs17116117	
Hypertension				
0	rs41346947	rs17046143	rs17008234	rs11005510
0	rs6840033	rs12525412		
0	rs10494787	rs11005510		
0	rs12525412	rs41472845	rs2865199	
0	rs10494787	rs41333648	rs825487	
0	rs17046143	rs16916476		
Rheumatoid Arthritis				
0	rs6620928	rs17222293		
0	rs1733717	rs5952117		
0	rs12839253	rs5952117		
0	rs41348644	rs5917288	rs10284137	
0	rs2782240	rs2412097		
0	rs4636358	rs7473249		
Type 1 Diabetes				
0	rs2596517	rs3135342		
0	rs13126272	rs13200022	rs11539216	
0	rs17055224	rs9310799	rs13126272	rs17808723
0	rs41385948	rs13126272	rs204990	
0	rs3094703	rs41445846		
0	rs6588238	rs1051336	rs6044514	
Type 2 Diabetes				
0	rs9842727	rs6421871	rs2416472	
0	rs16827563	rs10492267		
0	rs6842409	rs10492267		
0	rs2416472	rs17141753		
0	rs1890870	rs461043	rs17117531	
0	rs11196208	rs305040	rs17117531	

**Table 4 genes-16-01114-t004:** Subset of gene pairs involved in epistatic interactions identified by GPBSO across seven WTCCC GWAS datasets (complete list in Appendix A).

Gene 1	Ctd 1	Gene 2	Ctd 2	Number of Occurrences
Bipolar Disorder				
*COL19A1*	NDE	*LRIG1*	NDE	136
*MACROD2*	NDE	*NELL1*	NDE	44
*CLSTN2*	NDE	*EYA3*	NDE	27
*FAH*	NDE	*POU2F3*	NDE	25
*AOC1*	NDE	*FSTL5*	DE	20
*CA12*	NDE	*GPC5*	NDE	18
Coronary Artery Disease				
*CTNNA3*	NDE	*LRRTM3*	NDE	91
*DMD*	NDE	*STAG2*	NDE	33
*PUDP*	NDE	*RERE*	NDE	31
*GUCY2F*	NDE	*SCNN1B*	NDE	29
*GFOD1*	DE	*NLGN4X*	NDE	12
*GUCY1A1*	DE	*MIR99AHG*	NDE	4
Crohn’s Disease				
*DAB1*	NDE	*IL23R*	DE	169
*ERBB4*	NDE	*PTPRD*	NDE	147
*RBM47*	NDE	*WDFY3*	NDE	132
*ATG16L1*	DE	*RFTN1*	NDE	54
*LEP*	DE	*MGLL*	NDE	29
*NOD2*	DE	*STK10*	NDE	14
Hypertension				
*DNAH7*	NDE	*MIER1*	NDE	178
*SMOC2*	NDE	*STEAP1B*	NDE	82
*FAT4*	NDE	*KDM6A*	NDE	61
*PLSCR4*	NDE	*PYGL*	NDE	59
*ACE2*	DE	*PLD5*	NDE	2
*TGFA*	DE	*ZNF652*	NDE	2
Rheumatoid Arthritis				
*PRRC2A*	NDE	*SNORA38*	NDE	71
*CFB*	NDE	*NELFE*	NDE	68
*STS*	DE	*WDR53*	NDE	10
*ABCC5*	DE	*MSH5*	NDE	9
*ABCC4*	DE	*PTCHD1-AS*	NDE	9
*NR4A3*	DE	*TYK2*	DE	2
Type 1 Diabetes				
*LRIG1*	NDE	*PHTF1*	NDE	86
*MAGI3*	NDE	*NOTCH4*	NDE	45
*GTF2H4*	NDE	*TSBP1*	NDE	42
*HES1*	NDE	*TAP2*	NDE	28
*CD226*	NDE	*GPAT4*	NDE	19
*GPSM3*	NDE	*PTPN22*	DE	6
Type 2 Diabetes				
*MCF2*	NDE	*RAD51B*	NDE	43
*DNAJC2*	NDE	*PMPCB*	NDE	37
*DAB1*	NDE	*HGF*	NDE	36
*CCDC12*	NDE	*PRR16*	NDE	25
*GRM7*	NDE	*SLC25A15*	NDE	19
*LRMDA*	NDE	*TCF7L2*	DE	4

**Table 5 genes-16-01114-t005:** Representative genes participating in epistatic interactions detected by GPBSO across seven WTCCC GWAS datasets (complete list in Appendix A).

Gene	Ctd	Number of Occurrences
Bipolar Disorder		
*LRIG1*	NDE	260
*FSTL5*	DE	204
*COL19A1*	NDE	145
*RERE*	NDE	96
*EYA3*	NDE	85
*NTNG2*	DE	2
Coronary Artery Disease		
*DMD*	NDE	647
*FRMPD4*	NDE	455
*PTCHD1-AS*	NDE	308
*GFOD1*	DE	50
*GUCY1A1*	DE	4
*MRAS*	DE	3
Crohn’s Disease		
*RFTN1*	NDE	669
*IL23R*	DE	658
*DAB1*	NDE	558
*ERBB4*	NDE	506
*ATG16L1*	DE	108
*NOD2*	DE	73
Hypertension		
*MIER1*	NDE	195
*DNAH7*	NDE	183
*PIK3R3*	NDE	128
*P3R3URF-PIK3R3*	NDE	128
*TGFA*	DE	6
*ACE2*	DE	4
Rheumatoid Arthritis		
*TSBP1*	NDE	1057
*PTCHD1-AS*	NDE	712
*DMD*	NDE	675
*NR4A3*	DE	115
*STS*	DE	50
*ABCC5*	DE	15
Type 1 Diabetes		
*PHTF1*	NDE	119
*TSBP1*	NDE	107
*LY6G5B*	NDE	95
*LRIG1*	NDE	89
*TAP2*	NDE	74
*PTPN22*	DE	6
Type 2 Diabetes		
*CCDC12*	NDE	85
*HGF*	NDE	71
*RAD51B*	NDE	61
*MCF2*	NDE	54
*TCF7L2*	DE	14
*GLIS3*	DE	2

## Data Availability

Using the predefined penetrance tables, EpiGEN generated all three simulated datasets. The simulation datasets used in this study are available from the corresponding author upon reasonable request. For data used for the real data application in this project. Access to the WTCCC data can be requested from the owners via the following links: https://www.wtccc.org.uk/info/access_to_data_samples.html (access on 31 August 2025).

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
