# Peer review of "GPBSO: Gene Pool-Based Brain Storm Optimization for SNP Epistasis Detection"

_genes, 2025, doi:10.3390/genes16091114_

Round 1
Reviewer 1 Report
Comments and Suggestions for Authors
The manuscript presents GPBSO, a novel gene pool-based Brain Storm Optimization method for detecting high-order SNP epistasis in GWAS data. The work is innovative, well-motivated, and supported by strong validation on both simulated and real datasets. The results demonstrate clear advantages over existing methods, particularly in detecting third-order interactions. There are some strength of the paper
-
Innovative integration of swarm intelligence with a gene pool mechanism.
-
Thorough methodological exposition with clear diagrams and pseudocode.
-
Comprehensive experimental evaluation, including both synthetic and real GWAS datasets.
-
Biological relevance demonstrated by identifying meaningful gene–gene interactions.
However, the following questions should be addressed.
-
The computational cost of GPBSO compared to alternatives should be analyzed in more detail, possibly with runtime benchmarks.
-
More practical guidance on parameter tuning would improve usability for other researchers.
-
A deeper discussion on biological interpretability of the novel gene findings would strengthen the real-data application section.
Author Response
Comment 1: The computational cost of GPBSO compared to alternatives should be analyzed in more detail, possibly with runtime benchmarks. Response 1: Thank you for this insightful suggestion. We agree that computational cost is an important consideration. Therefore, we have added a new subsection entitled Computational Cost Analysis in the Results section (page 11), where we compare the runtime of GPBSO with other representative algorithms (DECMDR, SNPHarvester, AntEpiSeeker, HS-MMGKG, SEE). The new Table 2 presents average runtimes (in minutes) under consistent experimental settings. These results demonstrate that although GPBSO is slightly slower than filtering-based methods, it achieves comparable or superior efficiency relative to other swarm intelligence and evolutionary algorithms, while maintaining higher detection power. [Updated text has been inserted in the manuscript.] Comment 2: More practical guidance on parameter tuning would improve usability for other researchers. Response 2: We thank the reviewer for this valuable comment. To improve reproducibility and usability, we have added a new subsection entitled Parameter Settings and Practical Guidance in the Materials and Methods section (page 9). In this subsection, we provide recommendations on setting population size (PopSize), maximum generations (maxGen), and interaction order (mo), depending on dataset size and available computational resources. This practical guidance will help other researchers apply GPBSO effectively in different scenarios. [Updated text has been inserted in the manuscript.] Comment 3: A deeper discussion on biological interpretability of the novel gene findings would strengthen the real-data application section. Response 3: We appreciate this important suggestion. Accordingly, we have revised the Results (pages 13) and Discussion (page 17) sections to provide a deeper biological interpretation of novel candidate genes identified by GPBSO, including LRIG1, OMA1, and SPATA5. Supporting evidence from the Comparative Toxicogenomics Database (CTD) and relevant literature has been incorporated to strengthen the biological plausibility of these findings. [Updated text has been inserted in the manuscript.]Reviewer 2 Report
Comments and Suggestions for Authors
The manuscript requires extensive revision to make it understandable to readers who are not familiar with the system and the type of data.
In particular:
- The authors did not specify, in the heading preceding the title, which category the manuscript should be assigned to (see: "Type of the Paper (Article, Review, Communication, etc.) ". A comprehensive evaluation of the manuscript can only be made if it is clear which category the authors intend to submit it to.
- The graphic quality of the images must be improved because they need to be enlarged considerably, and under such conditions the quality is poor.
- Tables and figures provide little information on captions and content, and the presence of abbreviations and acronyms without any explanation makes them difficult to understand.
- I cannot find an explanation for Figure 4 to explain the inconsistency of the six panels, some of which contain six box plots and others four. The panels should be numbered or labelled with a letter, and a legend should provide information on the individual panels.
- the numbered list of streghts of the method in the introduction, similar to a description produced by a chatbot, is not appropriate in a descriptive introduction to the state of the art; it should be presented under Discussion, better in a different form, to outline the strengths of the method.
- the Discussion paragraph is very short and discusses poorly the results, just mentioning some weak point to be improved. The section should reports more in details strengths and weaknesses, I suggest to include a table that compares the characteristics of the new proposed method with the well-established methods used for comparison.
Author Response
Comment 1: The authors did not specify, in the heading preceding the title, which category the manuscript should be assigned to (see: "Type of the Paper (Article, Review, Communication, etc.) ". A comprehensive evaluation of the manuscript can only be made if it is clear which category the authors intend to submit it to. Response 1: Thank you for this reminder. We have clarified the manuscript type as Article in the heading preceding the title. Comment 2: The graphic quality of the images must be improved because they need to be enlarged considerably, and under such conditions the quality is poor. Response 2: We appreciate the reviewer’s comment. The figures have been regenerated at higher resolution (300 dpi) and re-inserted into the manuscript to ensure that they can be enlarged without loss of quality. We note that due to the higher resolution, scrolling through the document in Word may occasionally feel less smooth, but this ensures that the figures meet publication standards and remain clear after enlargement. Comment 3: Tables and figures provide little information on captions and content, and the presence of abbreviations and acronyms without any explanation makes them difficult to understand. Response 3: Thank you for pointing this out. We have expanded the captions of all tables and figures to provide clearer descriptions of their content. In addition, all abbreviations and acronyms are now defined at their first occurrence in the text, and key abbreviations are also explained within figure and table captions where necessary (e.g., SNP, GWAS, PopSize, maxGen, CTD, dbSNP, F-measure, and Power). Comment 4: I cannot find an explanation for Figure 4 to explain the inconsistency of the six panels, some of which contain six box plots and others four. The panels should be numbered or labelled with a letter, and a legend should provide information on the individual panels. Response 4: We thank the reviewer for this observation. Figure 4 has been revised: each panel is now labelled (a–f), and the figure legend has been expanded to explain the content of each panel and clarify the difference in the number of box plots. This provides a clearer and more consistent explanation of the results. Comment 5: The numbered list of strengths of the method in the introduction, similar to a description produced by a chatbot, is not appropriate in a descriptive introduction to the state of the art; it should be presented under Discussion, better in a different form, to outline the strengths of the method. Response 5: We appreciate this constructive suggestion. The numbered list of strengths has been removed from the Introduction. Instead, the strengths of GPBSO are now presented in the Discussion section in a more integrated and narrative form, highlighting the distinctive features of the method while avoiding a list-like style. Comment 6: The Discussion paragraph is very short and discusses poorly the results, just mentioning some weak point to be improved. The section should reports more in details strengths and weaknesses, I suggest to include a table that compares the characteristics of the new proposed method with the well-established methods used for comparison. Response 6: Thank you for this valuable recommendation. We have substantially revised and expanded the Discussion section to provide a more detailed analysis of both the strengths and limitations of GPBSO. Instead of a comparative table, we have summarized these points in narrative form to maintain a more cohesive flow in the discussion. This expanded text highlights GPBSO’s distinctive features relative to existing methods and provides a clearer and more comprehensive perspective.